# Attention 3D central difference convolutional dense network for hyperspectral image classification

**Mahmood Ashraf**[1], **Raed Alharthi**[2]\*, **Lihui Chen**[1]\*, **Muhammad Umer** [3], **Shtwai Alsubai**[4], **Ala Abdulmajid Eshmawi**[5]

**1** School of Micro Electronics & Communication Engineering, Chongqing University, Chongqing, China, **2** Department of Computer Science and Engineering, University of Hafr Al-Batin, Hafar Al-Batin, Saudi Arabia, **3** Department of Computer Science & Information Technology, The Islamia University of Bahawalpur, Bahawalpur, Pakistan, **4** Department of Computer Science, College of Computer Engineering and Sciences, Prince Sattam bin Abdulaziz University, Al-Kharj, Saudi Arabia, **5** Department of Cybersecurity, College of Computer Science and Engineering, University of Jeddah, Jeddah, Saudi Arabia

\* ralharthi@uhb.edu.sa (RA); lihui.chen@cqu.edu.cn (LC)

**Data Availability Statement:** All relevant data are present in the manuscript and Supporting information files. Data is also available via Github at https://github.com/mahmoodashraf8585/

## Abstract

Hyperspectral Images (HSI) classification is a challenging task due to a large number of spatial-spectral bands of images with high inter-similarity, extra variability classes, and complex region relationships, including overlapping and nested regions. Classification becomes a complex problem in remote sensing images like HSIs. Convolutional Neural Networks (CNNs) have gained popularity in addressing this challenge by focusing on HSI data classification. However, the performance of 2D-CNN methods heavily relies on spatial information, while 3D-CNN methods offer an alternative approach by considering both spectral and spatial information. Nonetheless, the computational complexity of 3D-CNN methods increases significantly due to the large capacity size and spectral dimensions. These methods also face difficulties in manipulating information from local intrinsic detailed patterns of feature maps and low-rank frequency feature tuning. To overcome these challenges and improve HSI classification performance, we propose an innovative approach called the Attention 3D Central Difference Convolutional Dense Network (3D-CDC Attention DenseNet). Our 3D-CDC method leverages the manipulation of local intrinsic detailed patterns in the spatial-spectral features maps, utilizing pixel-wise concatenation and spatial attention mechanism within a dense strategy to incorporate low-rank frequency features and guide the feature tuning. Experimental results on benchmark datasets such as Pavia University, Houston 2018, and Indian Pines demonstrate the superiority of our method compared to other HSI classification methods, including state-of-the-art techniques. The proposed method achieved 97.93% overall accuracy on the Houston-2018, 99.89% on Pavia University, and 99.38% on the Indian Pines dataset with the 25 × 25 window size.

Attention-3D-Central-Difference-Convolutional-Dense-Network-for-Hyperspectral-Image-Classification.

**Funding:** The author(s) received no specific funding for this work.;

**Competing interests:** The authors have declared that no competing interests exist.

# 1 Introduction

Hyperspectral Images (HS images) consist of numerous contiguous bands that provide extensive spectral information containing high spectral resolution features. HS images have various real-world applications, including urban analysis, land cover analysis, agriculture, and environmental analysis. HS image classification is an effective way to distinguish the variety of features and help in critical decision-making. Usually, the classification of these images relies on analyzing and interpreting the unique spectral signatures exhibited by the objects within them. However, real-world applications of hyperspectral imaging often face challenges such as low spatial resolution [1] caused by signal-noise ratio, sensor limitations, and complexity constraints. In past decades to address this, various classification techniques such as K-Nearest Neighbor (KNN) [2], Support Vector Machine (SVM) [3], Maximum Likelihood (ML) [4], Logistic Regression (LR) [2], and Extreme Learning Machine (ELM) [5] have been employed to classify spectral features, aiming for improved accuracy and robustness. Despite using these classifiers, their effectiveness is limited by redundant factors and high correlation among spectral bands, leading to suboptimal results. Furthermore, these classifiers exhibit suboptimal performance outcomes due to their inability to consider the spatial heterogeneity of hyperspectral image data. Achieving optimal classification accuracy requires the development of a classifier that effectively integrates both spectral and spatial information. The Spatial characteristics offer additional distinguishing details regarding an object's dimensions, configuration, and arrangement. Proper integration of these details can lead to more effective outcomes. The spatial-spectral characteristics of two distinct groups involve thoroughly analyzing their multifaceted properties and complex interdependencies. **1)** The analysis focuses on spatial and spectral features separately, conducting independent evaluations. Spatial attributes are obtained using advanced-level modules such as Entropy [5], Morphological [6, 7], Low-Rank Representation [8], and Attribute Self-Represent [9]. These spatial features are later merged with spectral features for pixel-level classification operations. **2)** The joint spectral-spatial features are investigated [10]. This involves the extraction of features that combine both spectral and spatial information, accomplished through the generation of 3-D wavelets, scattering wavelets [11], and Gabor filters [12] at various frequencies and scales. The traditional feature extraction methods relied on shallow and handcrafted learning approaches, which come from expert knowledge [13], which may potentially limit the application's ability to achieve accurate classification.

Recently, deep learning-based methods have shown effectiveness in various applications, specifically in image classification and object detection, by identifying the low to high-level features, which enables precise classification. The high spatial resolution of the HSI data is organized into 3D cubes, capturing complex details and effectively maintaining correlations between spectral and spatial features. This process enhances feature extraction and classification outcomes. Among these models, the convolutional neural network (CNN) [14] has gained popularity due to its superior ability to classify features compared to manually designed ones. That's strategy applied in various image processing tasks, including image classification [15, 16], object detection [17], semantic segmentation [18], colon cancer classification [19], depth estimation [20], face anti-spoofing [21], and related domains. Advanced techniques within the field of deep learning have been suggested for dealing with the problem of shifting domains in hyperspectral imaging (HSI). The paper [22] introduces a new approach called LRR-Net for anomaly detection. LRR-Net is a baseline network that combines the low-rank representation (LRR) model with deep learning techniques. LRR-Net employs the ADMM optimizer to efficiently solve the LRR model and convert normal parameters into trainable ones, hence reducing the need for human adjustment. The paper [23] presents a novel and comprehensive

framework for remote sensing (RS) applications, specifically designed to overcome the limited emphasis on spectral data in visual representation learning. SpectralGPT is designed specifically for processing spectral remote sensing (RS) images, unlike conventional models that mainly focus on RGB images. It utilizes a 3D-generated pre-trained transformer (GPT) architecture. It distinguishes itself by providing support for images of diverse sizes, resolutions, and time series through incremental training. A spatial and spectral BERT is proposed in [24] utilizing the local and global features to improve the HSI classification.

Recent years have witnessed remarkable progress in hyperspectral image analysis through deep learning techniques. Deep learning-based methods have already shown effectiveness in extracting the semantic deep features from HS images using stacked layers architectures either with 2-D or 3D convolutional neural networks that allowed the customization of spatial features, as shown in several studies [25–27]. However, it's essential to acknowledge that the 2D approaches are widely incorporated to extract spatial features independently, which might limit the full utilization of spectral-spatial data present in hyperspectral images. Moreover, these methods are weak in providing detailed, complex information about the spatial and spectral dimensions of the HS images. Although CNN-based frameworks provide deep information but cannot extract local intrinsic detailed and low-frequency information, such information is necessary for accurate classification. To overcome this limitation, We proposed a novel 3DCDCN dense architecture that is equipped with a 3D attention mechanism for exploring the more appropriate features, while the proposed dense connection provides robustness to the architecture towards the finely detailed and low frequency features from the HSIs. The CDC strategy is used to explore the intrinsic detailed information. In conclusion, the main contributions to this article are as

1. A customized 2D to 3D CDCN modules proposed that utilize central difference convolutional network techniques that incorporate central difference into vanilla convolution to enhance its representational characteristics and improve its generalization capacity. This method combines intensity and gradient data to extract intricate patterns within the spatial-spectral data. This advanced method offers a higher level of robustness and adaptability when compared to traditional CNN methodologies.

2. A novel CDCN architecture is introduced, which is equipped with 3D attention capabilities within the basic architecture of CDCN, which provides the robustness of the proposed method and efficiently considers detailed intrinsic features during the classification task.

3. A Dense Network module is introduced in the architecture that employs pixel-wise channel concatenation techniques to extract low-rank frequency features from 3D-CDCN, as explained in reference [28]. The spatial attention mechanism fine-tunes the 3D feature maps, enabling the model to fully leverage the benefits of low-rank frequency features while minimizing data loss.

4. Besides evaluating the effectiveness of the proposed CDCN architecture in terms of Overall accuracy (OA), Average accuracy (AA), and kappa coefficient (Kappa), we compared the efficiency of the proposed method with existing HSI classification-based methods.

This paper is organized into various sections such as Section 2 contains the related work while the methodology is discussed in Section 3, encompassing its technical aspects and theoretical foundations. Section 4 presents a comprehensive examination of the experimental datasets, accompanied by the analysis of results and discussions. Section 4.3 presents the ablation study. Section 5 indicates conclusions, emphasizing the findings and their implications for future research. The abbreviations and meanings are shown in Table 1.

**Table 1. Abbreviations and meanings.**

| Abbreviation | Meaning |
|---|---|
| 3D-CDC | 3D central difference convolution |
| 3D-CDCN | 3D central difference convolutional neural network |
| CDCN | central difference convolutional neural network |
| HSI | Hyperspectral image classification |
| HS image | Hyperspectral images |
| SA | Spatial Attention |
| CNN | convolutional neural network |

## 2 Related work

Recently, deep learning techniques for hyperspectral image (HSI) classification have been extensively explored. Various methods like Stacked Autoencoders (SAEs) [29], Deep Belief Networks (DBNs) [30], Deep Boltzmann Machines (DBMs) [31], and Convolutional Neural Networks (CNNs) [32–37] have garnered significant attention. SAE [29], for instance, functions as an unsupervised feature extraction technique by sequentially stacking autoencoders, allowing for the extraction of spectral-spatial features.

There are various applied domains of computer vision and digital image processing such as object detection, remote sensing image classification, medical image classification, video analysis, crime detection and industrial automation [38, 39]. Feature fusion and deep learning algorithms have shown robust results in various domains of computer vision [40, 41]. CNNs have seen widespread use in HSI classification. For instance, a basic CNN model with five layers was proposed [32], focusing solely on spectral information. To address this limitation, an enhanced CNN model [36] was introduced, utilizing 3-D patches as input to incorporate both spectral and spatial information. Furthermore, another approach involves a CNN integrated with a spatial pyramid pooling strategy to contain spatial information [36] comprehensively. Additionally, there's a proposition that combines CNN features with hand-crafted features and Conditional Random Field (CRF) [42]. Another variant, CNN with Markov Random Field (MRF) [33], was introduced to leverage label correlations effectively.

A dual-channel CNN [43, 44] was introduced, utilizing 1-D and 2-D CNNs for feature extraction. To expand the training dataset for deep CNNs, a novel pixel-pair method [37] was proposed. Moreover, a 3-D Convolutional Neural Network (3D-CNN) [45] was introduced, enabling joint spectral and spatial information processing. Similarly, a 3-D Contextual deep CNN (3D-FCN) [35] was suggested to optimize the exploration of local contextual interactions among neighboring individual pixel vectors. When we talk about applications of computer vision, then there are many research works done so far like on wheat classification [46], brightness correction [47], pattern analysis [48, 49], and photo-synthesis [50]. The transformer learning models also perform well for target object detection [51–53]. The advancement of Convolutional Neural Networks (CNNs) has spurred the development of various convolution techniques. One such technique, tiled convolution, employs distinct filters for feature map neurons with nearby receptive fields on the input image [54]. Consequently, this method generates a feature map using multiple filters, extracting more definitive features with an equal number of feature maps. In a study, the augmented linear mixing model (ALMM) tackles spectral variability in hyperspectral images by using a data-driven approach to isolate scaling factors linked to illumination or typography using an endmember dictionary, while also capturing additional variations from environmental factors

and instrument settings via spectral variability dictionary [55]. Their proposed method, integrated into the spectral unmixing framework, allows for the concurrent acquisition of the spectral variability dictionary and the estimation of abundance maps, showcasing enhanced effectiveness compared to earlier advanced techniques in experiments conducted on both synthetic and real datasets.

Dilated convolution [36], another convolutional approach, focuses on broadening the receptive field of a filter without increasing the parameters. This is achieved by introducing cells with zero weight values into the filters. Studies have indicated that this technique may enhance performance in certain scenarios [36].

In a different study, micro Multi-Layer Perceptron (MLP) structures are utilized as filters, referred to as Network In Network (NIN) [56, 57]. This enables the filters to learn more intricate relationships during the training phase. Another notable alternative convolution technique is Inception [58], where varying-sized filters are incorporated within a single convolution layer. This method can simultaneously execute convolution and pooling processes, demonstrating improved performance without escalating the parameter count by utilizing the inception module. While deep learning methods have accomplished impressive performance in HSI classification, they often require more data as input. Moreover, unlike traditional descriptors, Convolutional Neural Networks (CNNs) tend to overfit easily and face challenges in generalizing well to unseen scenes. This difficulty in generalization can pose a significant issue when applying CNNs for HSIs, hindering their adaptability to diverse and unfamiliar environments or scenes. Additionally, the reliance of these methods on extensive sequences as input poses practical limitations, particularly in scenarios where real-time processing is necessary. Therefore, despite achieving state-of-the-art results, these drawbacks highlight the need to improve the generalization ability and adaptability of CNN-based approaches in HSI classification tasks. The convolution operator plays a vital role in extracting fundamental visual features within the deep learning framework.

Recently, there have been developments and extensions to the conventional convolution operator. One direction involves incorporating classical local descriptors like LBP [59] and Gabor filters [60] into convolution design. Notable works include Local Binary Convolution [61] and Gabor Convolution [62, 63]. These innovations aim to save computational resources and enhance resistance to spatial changes. For instance, Local Binary Convolution is devised to reduce computational costs, while Gabor Convolution aims to improve resilience against spatial alterations. Another direction in extending convolution operators involves modifying the spatial scope for aggregation. Noteworthy works in this area include dilated convolution [64] and deformable convolution [65, 66]. These adaptations aim to alter the convolution's receptive field, allowing for wider spatial information aggregation.

However, these convolutional operators have primarily been designed and studied for the RGB modality. How effectively they perform when applied to depth and abundant spectral modalities remains uncertain. Understanding their performance across different modalities, such as depth and Hyperspectral data, requires further exploration and investigation. Specifically, how these modified convolutional techniques function and adapt to HSI data is an open question that must be addressed to understand their efficacy across various modalities comprehensively. These alternative techniques aim to increase the number of acquired features from a sole convolution layer. Aligning with this notion, this paper introduces a novel convolution technique termed "central difference convolution." The proposed method harbors unique qualities and advantages compared to existing approaches.

## 3 Proposed methodology

The given assumption is that the Hyperspectral Image, which consists of spectral-spatial features, can be represented as follows:

$$\mathrm{X} = [x_{n1}, x_{n2}, x_{n3}, \cdots, x_{nL}]^T \in \mathbb{R}^{(L_B \times (H \times W))} \quad (1)$$

The dataset comprises multiple $L$ bands, with each band containing H × W samples. Within each band, there are $C_L$ classes assigned to every sample. Here, $X$ denotes the original input, with $L_B$ denoting the number of spectral bands, $W$ representing the width, and $H$ representing the height. The input $X$ comprises Hyperspectral pixels with $L_B$ spectral measurements, which are utilized to generate a one-hot label vector for each pixel as follows:

$$\mathrm{X} = [y_{n1}, y_{n2}, y_{n3}, \cdots, y_{nC}] \in \mathbb{R}^{(1 \times 1 \times C_L)} \quad (2)$$

The classification of Hyperspectral pixels in land cover categories, denoted as $C_L$, poses a significant challenge for classification models. This complexity arises from various factors, such as diverse land-cover classes, inter-class similarity in heights, intra-class variability in heights, and overlapping and nested regions. Overcoming these complexities requires substantial and intensive efforts [67–71]. Consequently, any model aiming to address and resolve these issues effectively faces significant obstacles. To mitigate the dimensionality of spectral bands from $L_B$ to $B$ while preserving spatial features in terms of height ($H$) and width ($W$), the Principle Component Analysis ($PCA$) method is employed. This method is illustrated in Fig 1, where selective reduction of spectral bands retains critical spatial feature information essential for object recognition. Following the application of $PCA$, the data cube is transformed into a modified input $X \in \mathbb{R}^{(W \times H \times B)}$, where $W$ represents width, $H$ represents height, and $B$ represents the number of spectral bands retained after $PCA$ reduction. The data cube features are divided into small-scale, overlapping 3-D patches for HSI classification. The ground truth labels for these patches are determined based on the pre-defined label of the central pixel, facilitating accurate classification of the entire data cube. When generating $P \in \mathbb{R}^{(S_{ws} \times S_{ws} \times B)}$, we use $X$ inputs. $P$ represents a set of 3-D neighboring patches, and each patch is centered at the spatial location $(\alpha, \beta)$, covering a spatial extent of $S_{ws} \times S_{ws}$, where $S_{ws}$ is the window size. The parameter $B$ represents the number of spectral bands. The number of 3-D neighboring patches generated from the set $X$ depends on $(W - S_{ws} + 1) \times (H - S_{WS} + 1)$, where $W$ and $H$ are the dimensions of the original data matrix, and $S_{ws}$ is the patch size in each dimension. Therefore, the 3D neighboring patches at location $(\alpha, \beta)$, denoted as $P_{(\alpha, \beta)}$, are characterized by their distinctive features, covering a range from $(\alpha - (S_{ws} - 1)/2)$ to $(\alpha + (S_{ws} - 1)/2)$ in width and from $(\beta - (S_{ws} - 1)/2)$ to $(\beta + (S_{ws} - 1)/2)$ in height, inclusive of all $B$ spectral bands present in the PCA-reduced data cube $X$ features. Then 3D cubic patches are forwarded to the CDCN blocks, where these blocks are arranged in specific patterns as presented in Fig 1, where each block is composed of two CDCN modules and one 3D attention module which are placed between these two CDCN blocks, on the one hand, this provides the novel architecture with 3D Attention and on the other providing the robustness to the architecture and helps to minimize the computational complexity with more accurate classification accuracy. The output of Block 1, Block 2, and Block 3 are concatenated and passed from the flattened layers. To deeply extract the features, these features are passed further from two more CDCN blocks, and at the end, features are passed from the fully connected layer for the classification task. More detail about the composition of the block and the Denseness of the proposed network is discussed in the next section.

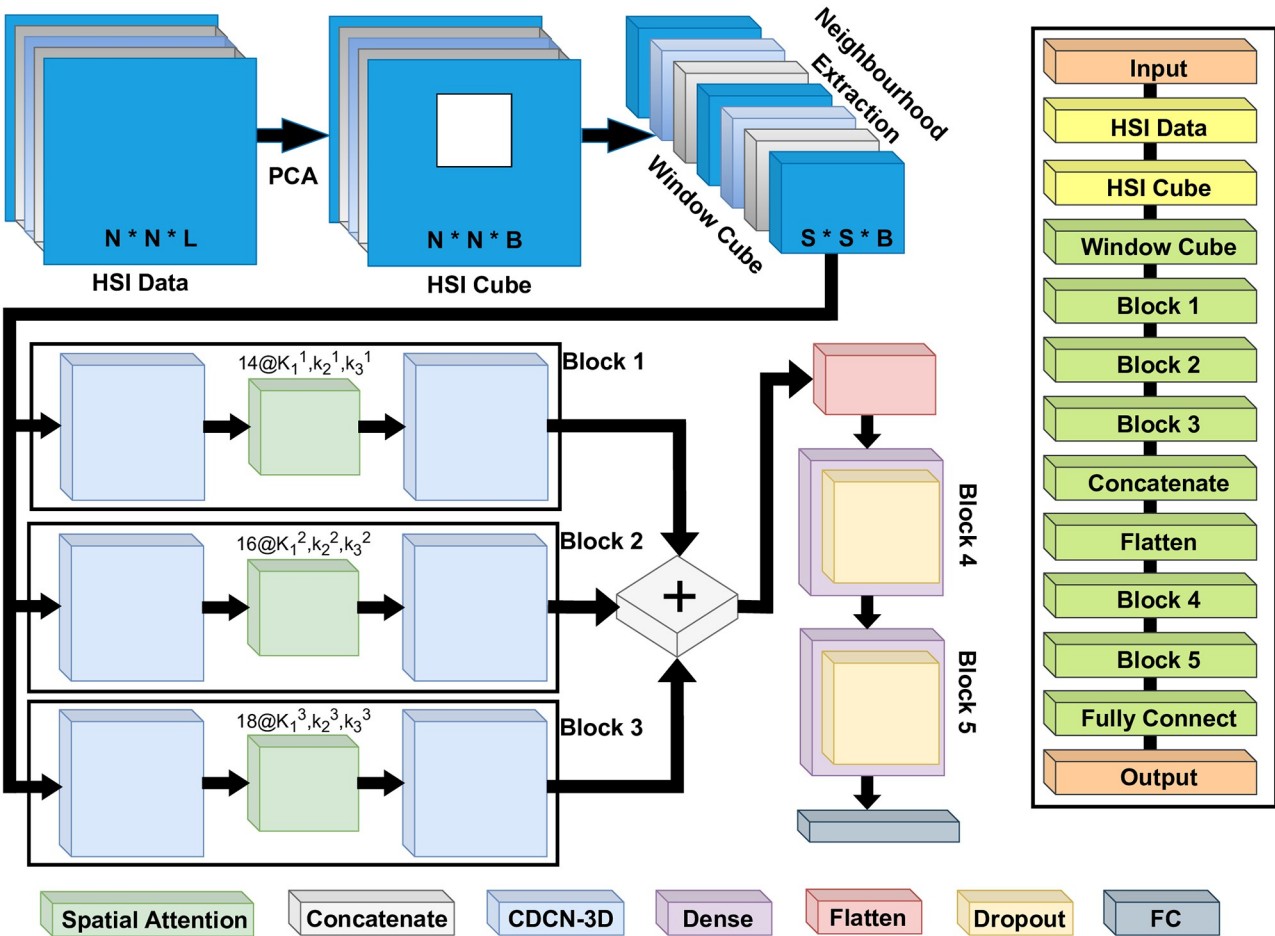

**Fig 1. The attention 3D-CDC dense network combines the 3D-central difference convolutional and attention methods to create an integrated architecture for classifying hyperspectral images.**

## 3.1 Central difference convolution

The CDC, as utilized in the proposed method, comprises a convolutional layer designed to extract features from hyperspectral data. Specifically, it leverages the concept of central differences within the convolutional operation to capture complex spectral-spatial information in hyperspectral images. The CDCN module computes the central differences between adjacent spectral bands or channels during the convolution process. Utilizing this technique, the module aims to enhance feature representation by emphasizing subtle spectral variations across neighboring bands. This allows for extracting discriminative features that encapsulate spectral and spatial characteristics unique to hyperspectral data. In modern deep learning frameworks, the conventional operators play a fundamental role in capturing spatial-spectral features. The convolution operation of 3D-CNN remains consistent across the channel dimension. The CDC approach is implemented with CNN to fully grasp the finely detailed information from the HSIs. This delicate information is vital in obtaining accurate classification. The following subsections will briefly explain how the CDC incorporated with CNN.

**3.1.1 Vanilla convolution.** The main operation utilized in Convolutional Neural Networks (CNNs) for visual tasks is known as the 3D spatial-spectral vanilla convolution. Referred to as vanilla convolution, it involves two key steps. The initial step entails selecting a local receptive field region denoted as $R_l$ from the input feature map $X_{fm}$. The subsequent step involves aggregating the sampled values by means of weighted summation using $W_{sv}$. As a result, the output feature map $Y_{fm}$ can be expressed as *CDC*.

$$Y(P_0) = \sum_{P_n \in R_l} W_{sv}(P_n) \cdot X_{fm}(P_0 + P_n) \tag{3}$$

In the context of coordinate representation in feature maps, $P_0$ denotes the current coordinates for both the input and output maps. On the other hand, $P_n$ serves as a variable that enumerates the coordinates within the region $R_l$. To provide an example, let's consider a 3D-CNN convolution operation with a kernel size of $3 \times 3$ and a dilation value of 1. In this case, the region $R_l = \{(-1, -1, -1), (-1, -1, 0), \ldots, (0, 1, 1), (1, 1, 1)\}$ corresponds to the specific local receptive field region.

**3.1.2 Vanilla convolution (Central difference).** The suggested methodology is based on the renowned local binary pattern [72, 73], which analyzes the intricate local relationships using a binary central difference approach. Our proposal incorporates central difference into vanilla convolution to enhance its representational characteristics and improve its generalization capacity. Central Difference Convolutional (CDC) consists of two essential and interconnected stages: Sampling and Aggregation. The sampling process resembles vanilla convolution, while the aggregation step differentiates itself through the method depicted in Fig 2. In CDC, we prioritize amalgamating the gradient of sample feature values directed towards the center. Eq 3 is modified accordingly.

$$Y(P_0) = \sum_{P_n \in R_l} W_{sv}(P_n) \cdot (X_{fm}(P_0 + P_n) - X_{fm}(P_0)) \tag{4}$$

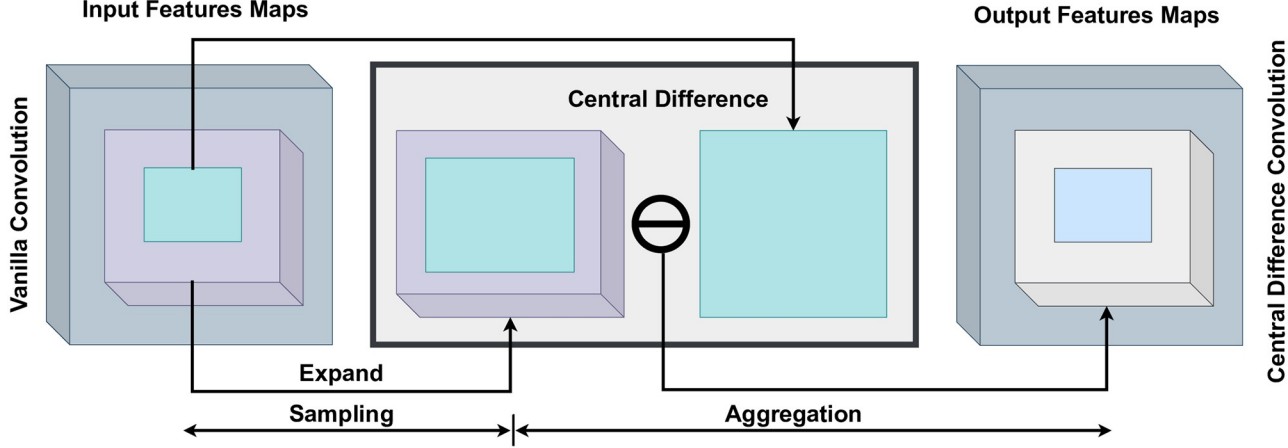

**Fig 2. The generalized method of Central Difference Convolution (CDC).**

At the origin $(0, 0, 0)$, the gradient sample feature value of $Pn$ is consistently zero in relation to the central location $P_0$. The classification of Hyperspectral Images involves analyzing complex and interconnected patterns, such as intensity-level semantics and gradient-level details, which are crucial and mutually supportive. To address this, combining the vanilla 3D-CNN convolution with 3D-CDC can be a promising method to improve the modeling capacity, offering increased resilience and discrimination. Therefore, we present a generalized formulation of central difference convolution as follows:

$$
\begin{aligned}
Y(P_0) \quad &= \theta_h \cdot \underbrace{\sum_{P_n \in R_l} W_{sw}(P_n) \cdot (X_{fm}(P_0 + P_n) - X_{fm}(P_0))}_{\text{CDC}} \\
&+ (1 - \theta_h) \cdot \underbrace{\sum_{P_n \in R_l} W_{sw}(P_n) \cdot X_{fm}(P_0 + P_n)}_{\text{VC}}
\end{aligned}
\tag{5}
$$

The hyper-parameter $\theta_h$ is crucial in balancing the impact of intensity-level semantic information and gradient-level detailed features information. This parameter is confined to the closed interval $[0, 1]$. Increasing the value of $\theta_h$ amplifies the significance of central difference gradient information, enhancing its overall contribution.

**3.1.3 Implementation for CDC.** To successfully integrate *CDC* into modern deep learning frameworks, we adopt a strategy that involves decomposing and combining Eq 5 by incorporating a vanilla convolution and an extra central difference term. This novel convolution technique, known as *CDC*, draws its name from a similar concept introduced in [74, 75]. By implementing this approach, we enhance the effectiveness of *CDC* within contemporary deep learning frameworks.

**3.1.4 Detailed derivation for CDC.** We have thoroughly examined the precise derivation of the Central Difference Convolution (CDC), which is a crucial element of our suggested model. This derivation seeks to clarify the mathematical foundations of the CDC, including its essential elements, the Central Difference Term (CDT) and Vanilla Convolution (VC). The elements stated combined provide the key basis of our method, as represented by Eq 6. This discussion aims to clarify the operational mechanics of the CDC, offering a clear

understanding of how it significantly improves the feature extraction capabilities of our model.

$$
\begin{aligned}
Y(P_0) \quad &= \theta_h \cdot \underbrace{\sum_{P_n \in R_l} W_{sw}(P_n) \cdot (X_{fm}(P_0 + P_n) - X_{fm}(P_0))}_{\text{CDC}} \\
&+ (1 - \theta_h) \cdot \underbrace{\sum_{P_n \in R_l} W_{sw}(P_n) \cdot X_{fm}(P_0 + P_n)}_{\text{VC}} \\
&= \theta_h \cdot \underbrace{\sum_{P_n \in R_l} W_{sw}(P_n) \cdot (X_{fm}(P_0 + P_n)}_{\text{VC}} \\
&+ \theta_h \cdot \underbrace{(-\sum_{P_n \in R_l} W_{sw}(P_n) \cdot (X_{fm}(P_0))}_{\text{CDT}} \\
&+ (1 - \theta_h) \cdot \underbrace{\sum_{P_n \in R_l} W_{sw}(P_n) \cdot (X_{fm}(P_0 + P_n)}_{\text{VC}} \\
&= (\theta_h + 1 - \theta_h) \cdot \underbrace{\sum_{P_n \in R_l} W_{sw}(P_n) \cdot (X_{fm}(P_0 + P_n)}_{\text{VC}} \\
&+ \theta_h \cdot \underbrace{(-X_{fm}(P_0) \cdot \sum_{P_n \in R_l} W_{sw}(P_n))}_{\text{CDT}} \\
&= \theta_h \cdot \underbrace{\sum_{P_n \in R_l} W_{sw}(P_n) \cdot (X_{fm}(P_0 + P_n)}_{\text{VC}} \\
&+ \theta_h \cdot \underbrace{(-X_{sw}(P_0) \cdot \sum_{P_n \in R_l} W_{sw}(P_n)}_{\text{CDT}}
\end{aligned}
\tag{6}
$$

In this equation, $Y(P_0)$ represents the output at the central pixel $P_0$. $\theta_h$ is a hyperparameter that balances the contributions of the central difference term (CDC) and the value convolution term (VC). $W_{sw}(P_n)$ denotes the spatial weights for neighboring pixels $P_n$ within the local region $R_l$. $X_{f}m(P_0 + P_n)$ and $X_{f}m(P_0)$ are the feature map values at the neighboring pixel $P_n$ and the central pixel $P_0$, respectively. The model combines the CDC and VC terms, where the CDC captures local differences in feature maps, emphasizing edges and fine details. At the same time, the VC aggregates local information, ensuring robustness to noise. The hyperparameter $\theta_h$ allows for the adjustment of the model's sensitivity to local differences.

## 3.2 Attention module

In Fig 3, we are presented with a visualization of the attention module, which is an integral part of the CDC Network. Following the initial step, the spatial module acquires the spatial $S$ map $M_S$ for the *CDCN* features, enabling a detailed process of refinement and fine-tuning.

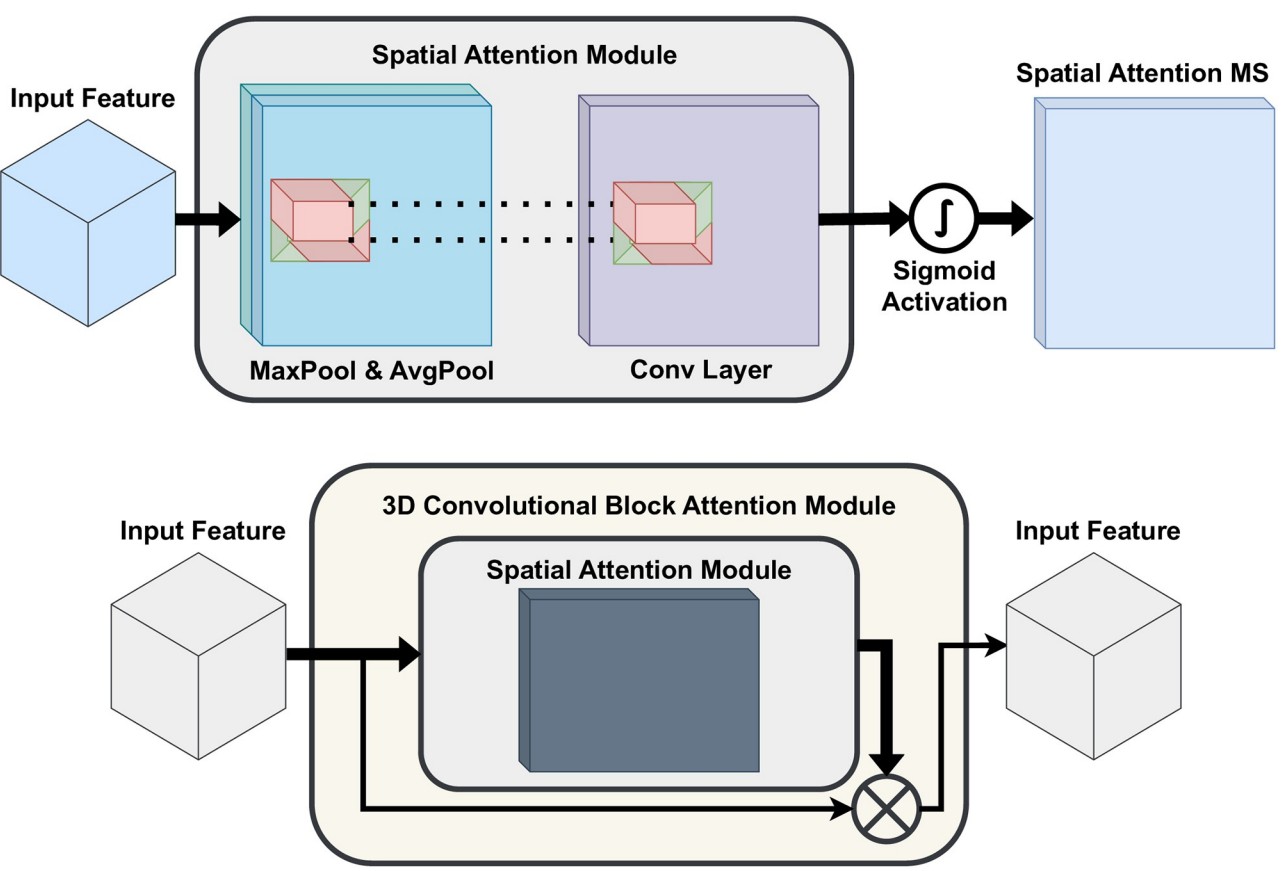

**Fig 3. Spatial attention module.**

This comprehensive methodology can be explained as follows:

$$F'_S = M_S(F) \otimes F \tag{7}$$

The attention block in natural image processing has demonstrated its empirical effectiveness by employing the $\otimes$ feature-wise multiplication method [76, 77]. This involves implementing average pooling $F^C_{avg}$ and max pooling $F^C_{max}$ operations along the filter axis, which primarily directs the spatial attention mechanism towards the *CDC* spatial feature maps *C*. The resulting descriptors from this concatenation process are then passed through the convolution function *f*. Subsequently, the output transforms the non-linear sigmoid activation function, represented as *σ*. To summarize, the entire procedure can be briefly described as follows:

$$F^{CDC}_S = \sigma(f[F^C_{avg}; F^C_{max}]) \tag{8}$$

### 3.3 3D-CDC Attention Dense Network model

The 3D-CDC Attention Dense Network model is exhaustively expounded by elucidating the various categories of layers employed, the dimensions of output maps generated, and the numerical count of parameters entailed. The specific layer types and their corresponding

parameters are represented in detail in Fig 4. It should be noted that the Input Layer dense layer contains the greatest number of parameters among all the layers. Additionally, it can be noted that the number of nodes present in the final dense layer corresponds to the number of categories present within the Houston2018 dataset, which is specifically seven. The proposed model's parameter count varies based on the number of classes within the dataset, thus rendering the determination of the total parameter count complex. Furthermore, it is worth mentioning that the network's weights undergo a random initialization process, followed by training using the back-propagation algorithm in conjunction with the Adam optimizer and utilizing the Softmax loss function. The training process uses mini-batches consisting of 256 units and is iterated throughout 100 epochs. It is imperative to mention that batch normalization and data augmentation techniques are not implemented during training.

## 4 Experiments and discussion

### 4.1 Datasets explanation

During the experimental phase of this study, the analysis included three Hyperspectral datasets: Indian-Pines (IP), University of Pavia (UP), and Houston-2018 (HT). Additionally, a comprehensive description of each Hyperspectral dataset was provided. These datasets are publicly available for the experiments and can be downloaded from the website www.ehu.eus [78].

**Indian Pines (IP):** In 1992, the Air-Borne Visible/Infrared Imaging Spectro-meter [79] sensor was utilized to procure the dataset recognized as IP. The area comprised several agricultural fields with an organized geometric structure and some areas of irregular forest. The image under analysis encompasses a vast $(145 \times 145)_P$ pixel array, containing an extensive collection of 224 spectral bands that span the wavelength range of 400 to 2500 nanometers, all of which are captured at a remarkable spatial resolution of 20 meters per pixel. After removing four null bands and 20 other bands affected by atmospheric water absorption, the pre-processed data consisted of 200 remaining bands utilized for experimentation. Additionally, almost 50% of the dataset, i.e., $(10, 249)_P$ pixels out of the total $(21, 025)_P$, contained ground-truth information that provided a single label belonging to one of the 16 different classes.

**Pavia University (UP):** The UP dataset, including the Northern Italian campus of UP, was obtained using the Reflective Optics System Imaging Spectro-meter [80] sensor. It mainly encompasses an urban environment characterized by numerous solid structures such as Asphalt, Gravel, and metal sheets, along with natural objects such as Trees, Meadows, and Soil. The dataset also includes shadows. After the elimination of the noisy bands, a total of 103 spectral bands were obtained within the spectral range of 0.43 to 0.86 meter, with a spatial resolution of $(1.3)_{MPP}$, and consisting of $(610 \times 340)_P$ pixels in size, each matched pixel by pixel. Out of the total of $(207, 400)_P$ pixels, a significant proportion comprising precisely 20%, amounting to $(42, 776)_P$ pixels, have been accurately annotated to contain authentic ground-truth information belonging to as many as nine distinctive class labels. The present circumstance entails the utilization of a model processing size measuring 200 units in width, 200 units in height, and 103 units in bands.

**Houston 2018 (HT):** The present study involves the analysis of a scene from Houston in 2018, which encompasses an area of 210 by 954 pixels, comprising a total of 48 spectral-bands within the wavelength range of 380 to 1050 nanometers, with 1 ground sample interval meter. The ground-truth data for the Houston 2018 scene has a pixel size of 0.5, meaning each pixel in the image corresponds to a physical area of 0.5 square units. To process the Houston 2018 scene, a model with dimensions of 200 units in width, 200 units in height, and 48 units in

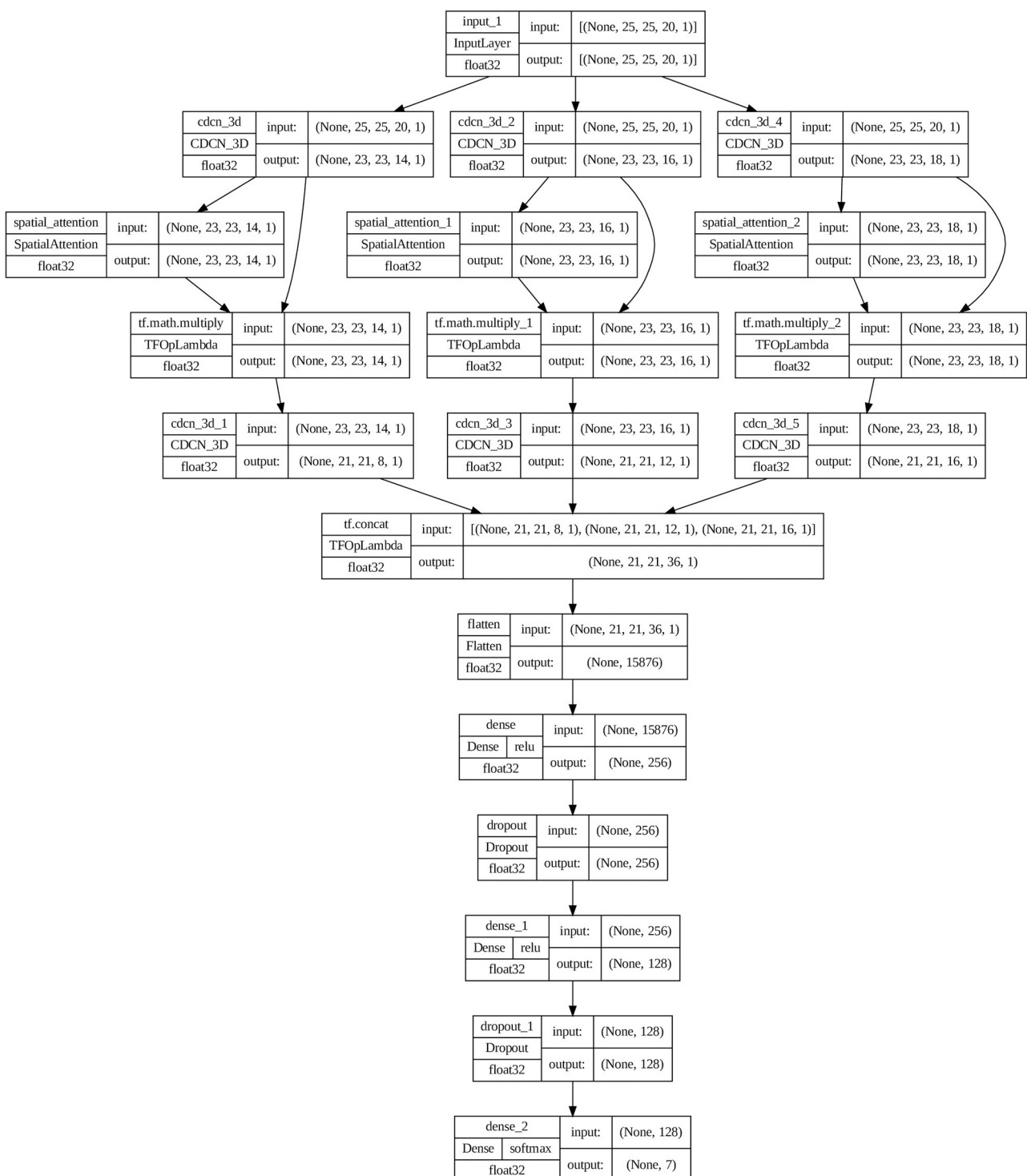

**Fig 4. The proposed 3D-CDC Attention Dense Network architecture is summarized layer-wise, employing a window size 25x25.** The final layer of this architecture is specifically designed using the Houston 2018 dataset.

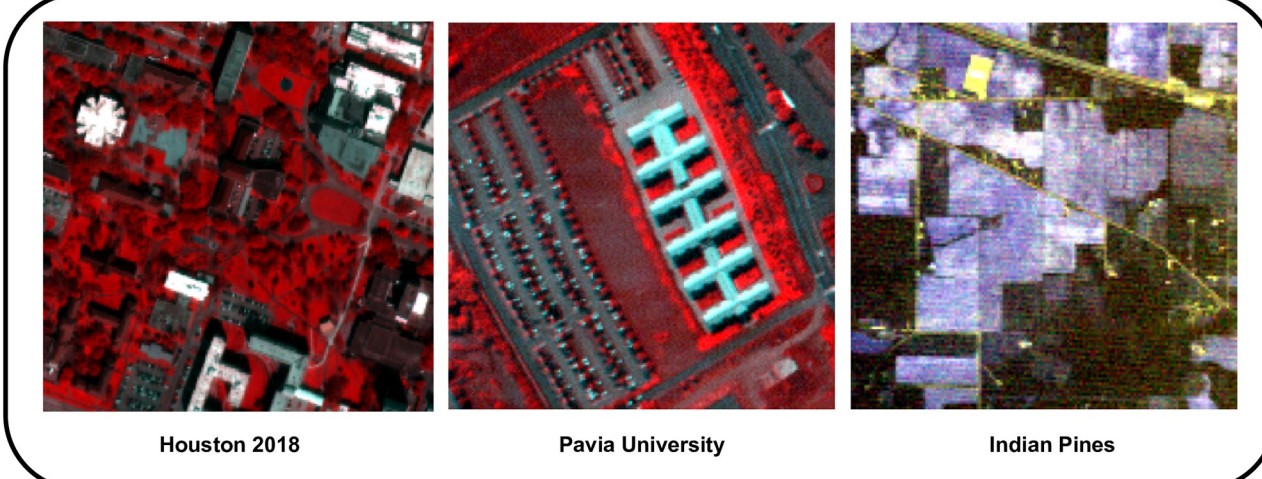

**Fig 5. The experimental datasets utilized in this study involved ground images, which were analyzed to derive insights and draw conclusions.**

bands is utilized. The reference [78] provides further details about the experimental datasets. Fig 5 presents visual representations of the ground images for all the experimental datasets.

**4.1.1 Experimental settings.** Experiments are done in this study using the proposed model and comparing methods. Three well-known HS HSI datasets were taken to check the efficiency of the 3DCDCN. Houston-2018 (HT), IP, and PU were used to test. All the experiments were done in Colab Pro.

In all experiments, the initial Test/Train set was divided into a 30: 70, ratio, with 70% of the population reserved for train samples and the remaining 30% for test samples. To ensure fair evaluations, we used a standardized learning rate of $(1e-02)$ and a decay rate of $(1e-06)$ for all experiments. Furthermore, we utilized the rectified linear unit activation function in all layers except for the last layer, where we employed the softmax activation. The patch sizes used in our experiments were $13 \times 13 \times 20$, $17 \times 17 \times 20$, $21 \times 21 \times 20$, and $25 \times 25 \times 20$, respectively. These patch sizes were determined using the PCA method to identify the 20 most informative bands. For optimization, the Adam optimizer was used, and 100 epochs were set to train the model.

**4.1.2 Evaluation metrics.** The evaluation of hyperspectral image (HSI) classification performance requires the use of multiple assessment metrics, namely overall accuracy (OA), average accuracy (AA), and kappa coefficient (Kappa), which are interconnected and intricate. *OA* measures the proportion of correctly classified test samples among the total testing samples, calculated by the given equation.

$$OA = \frac{\text{No. of correct predictions}(\sum_i X_{test})}{\text{Total number of predictions}} \tag{9}$$

*AA* calculates the average accuracy across different classes. The mathematical equation of the *AA* is given by

$$AA = \frac{1}{\text{No. of classes}} \sum_{i=1}^{n} \frac{\sum_{j=1}^{n} X_i^j}{T_i} \tag{10}$$

*Kappa* is a statistical metric that quantifies the agreement between the ground truth and the classification map and is calculated by the equation.

$$k = \frac{(A_o - A_e)}{1 - A_e} \qquad (11)$$

A comprehensive analysis of these evaluation measures is essential for accurately assessing HSI classification performance.

**4.1.3 Experiments of proposed method using the different window sizes.**   First, we measured the efficiency of the proposed model on the different window sizes, and the achieved outcomes are listed in Table 2. For our experiments, four cases were formulated based on different-sized windows to evaluate the efficiency of the proposed model.

**Case 1:** In the first scenario, we selected the patch size $13 \times 13$ using the 20 spectral bands. Experiments were performed on the three datasets, i.e., HT, PU, and IP. The proposed method achieved the OA 97.75% with the HT dataset, 99.57% on the PU, and 98.94% on the IP dataset. The proposed method outperformed the IP dataset. When we select the window size $13 \times 13$.

**Case 2:** In the second case, we selected the window size $17 \times 17$ and noted the efficiency of our method. In Table 2, it can be seen that 3D-CDC Attention DenseNet produced the highest 99.80% OA on PU images

**Case 3:** $21 \times 21$ window size was used to check the behavior of 3DCDCN ATT Dense Net on the under-study HSI datasets. With this window size, the proposed method achieved the top overall accuracy on PU, i.e., 99.86%, and the second best produced on the IP dataset with 98.71%.

**Case 4:** In this case, the highest window size $25 \times 25$ was chosen for the experimental purposes. The proposed method consistently showed effectiveness, achieved the highest OA on the PU dataset at 99.89%, and produced the second-best results with IP. Experiments using all these cases show that our method can perform well on different size window portions and produce good results with other datasets. Moreover, all the experimental settings utilized an equal number of spectral bands. i.e., 20. In the next part of the experiments, the performance of the proposed method will be compared with the benchmarks.

**4.1.4 Comparison with benchmarks.**   In comparative analysis, five main approaches use Convolutional Neural Networks (CNNs). These approaches include the Semi-Supervised 3D-CNN method [81], the Spectral-Spatial 3D-SSCNN method [82], the Fast and Compact 3D-FCCNN method [83], the Hybrid-SN method [13], and the Jigsaw-HSI method [84]. We

**Table 2. Based on our research findings, the efficacy of the proposed model is contingent upon the size of the window.**

| Datasets | HT | | | PU | | | IP | | |
|---|---|---|---|---|---|---|---|---|---|
| Window size | OA% | AA% | Kappa% | OA% | AA% | Kappa% | OA% | AA% | Kappa% |
| 13x13 | 97.75 | 76.01 | 92.29 | 99.57 | 87.97 | 99.46 | 98.94 | 98.48 | 98.79 |
| 17x17 | 97.78 | 74.28 | 92.54 | 99.80 | 99.41 | 99.75 | 98.94 | 99.16 | 98.79 |
| 21x21 | 97.82 | 73.80 | 92.44 | 99.86 | 99.83 | 99.82 | 98.71 | 98.50 | 98.53 |
| 25x25 | 97.93 | 71.74 | 92.80 | 99.89 | 88.82 | 99.86 | 99.38 | 99.45 | 99.30 |

performed extensive experiments using the proposed method and comparing methods to find the efficiency of our proposed method.

First, all the models were tested by selecting the window size 13 × 13. This is the small window size of our experiments. 20 number of spectral bands was used with all the window sizes. All comparing model results are listed in Table 3 and in Fig 6. The 3D-FCNN algorithm showed good results on the HT and PU datasets but could not maintain consistency on the IP dataset. It predicted an overall accuracy of 95.21%, which is less than 0.02 than the 3D-CNN model 3D-CNN provided better results on the IP dataset than the rest of the algorithms except the proposed model. In this series, Jigsaw could not show the performance on HT, PU, and IP datasets. In contrast, the proposed method showed superior results than the comparing methods and achieved OA = 98.75%, AA = 79.01%, Kappa = 92.29% on the HT dataset, and achieved OA = 99.57%, AA = 89.97%, Kappa = 99.46% on PU where as scored OA = 98.94%, AA = 98.48%, Kappa = 98.79% on the IP dataset which are the good results on this window size.

**Table 3. In light of the spatial dimension of the 13 × 13 window size, comparative evaluations were conducted with comparing methods.**

| Datasets | HT | | | PU | | | IP | | |
|---|---|---|---|---|---|---|---|---|---|
| Models | OA | AA | Kappa | OA | AA | Kappa | OA | AA | Kappa |
| 3D-CNN | 97.63 | 78.07 | 91.75 | 98.95 | 79.57 | 98.69 | 95.23 | 95.7 | 94.55 |
| 3D-SSCNN | 97.26 | 76.20 | 90.93 | 99.44 | 88.19 | 99.30 | 93.03 | 92.66 | 92.04 |
| 3D-FCCNN | 98.36 | 64.92 | 91.46 | 99.38 | 77.51 | 99.23 | 95.21 | 71.28 | 94.53 |
| HYbrid-SN | 95.36 | 47.46 | 84.06 | 99.18 | 77.01 | 98.98 | 76.82 | 50.08 | 72.91 |
| JIgsaw-HSI | 91.66 | 40.12 | 67.21 | 89.93 | 69.51 | 87.29 | 49.63 | 24.99 | 38.87 |
| Proposed | **98.75** | **79.01** | **92.29** | **99.57** | **89.97** | **99.46** | **98.94%** | **98.48** | **98.79** |

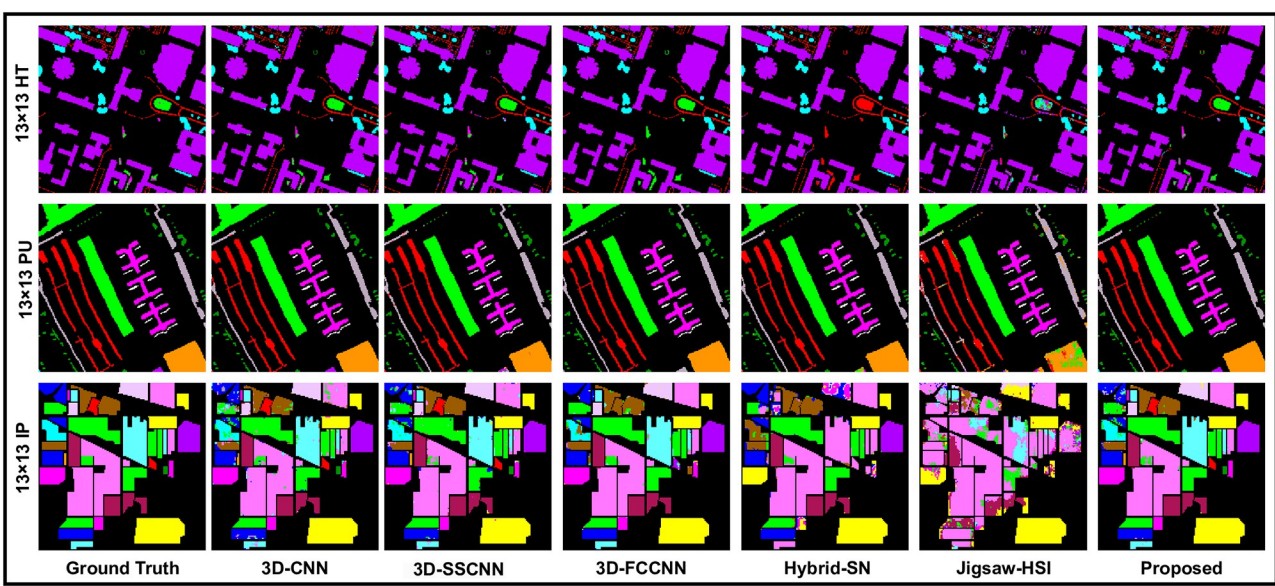

**Fig 6. Our models, which incorporate state-of-the-art techniques, perform sophisticated processing of the Ground Truths with high precision and accuracy, considering the complexity of each spatial dimension.**

**Table 4. In light of the spatial dimension of the 17 × 17 window size, comparative evaluations were conducted with cutting-edge deep learning techniques.**

| Datasets | HT | | | PU | | | IP | | |
|---|---|---|---|---|---|---|---|---|---|
| Models | OA | AA | Kappa | OA | AA | Kappa | OA | AA | Kappa |
| 3D-CNN | 96.23 | 75.67 | 86.17 | 99.03 | 76.89 | 98.79 | 95.30 | 92.34 | 94.63 |
| 3D-SSCNN | 96.73 | 71.78 | 89.28 | 99.57 | 88.32 | 99.46 | 94.67 | 94.54 | 93.91 |
| 3D-FCCNN | 95.69 | 48.26 | 85.31 | 99.41 | 77.66 | 99.27 | 96.30 | 72.45 | 95.77 |
| HYbrid-SN | 96.6 | 57.50 | 88.38 | 98.75 | 75.30 | 98.44 | 74.07 | 43.56 | 69.45 |
| JIgsaw-HSI | 94.51 | 53.38 | 79.79 | 92.32 | 72.54 | 72.54 | 62.04 | 37.61 | 55.37 |
| Proposed | **97.78** | **76.28** | **92.54** | **99.80** | **99.4** | **99.75 %** | **98.94** | **99.16** | **98.79** |

To further check the reliability of the proposed method against the comparing methods on different patch sizes, experiments were performed using the window size or patch size 17 × 17, where 3D-SSCNN showed satisfactory results on HT and PU dataset but provided the lower AA = 71.78% which is lower than 3D-CNN and predicted the lower OA on IP, which is lower than 3D-CNN and 3D-FCNN. HYbridSN and JIgsaw-HSI predicted the overall poor numeric values. Whereas the proposed methods predicted the overall top-class results on the measuring scales, these results are listed in Table 4 and in Fig 7.

When we came to the experiments with window size 21 × 21, The proposed 3D-CDC attention-dense net showed effectiveness and superior results. Meanwhile, Jigsaw-HSI improved the accuracy on the measuring scales but still produced the lowest prediction. 3D-FCCNN could only produce better OA (97.50%) on HT and showed unsatisfactory results on the remaining measuring scales and the as well as on the other two datasets. These outcomes are recorded in Table 5 and Fig 8. To make the experiments more comprehensive, a 25 × 25 patch size was also used where 3D-FCCNN predicted OA = 97.55%, which is the second highest result on HT, but AA was still lower than most of the methods, which placed this model in the

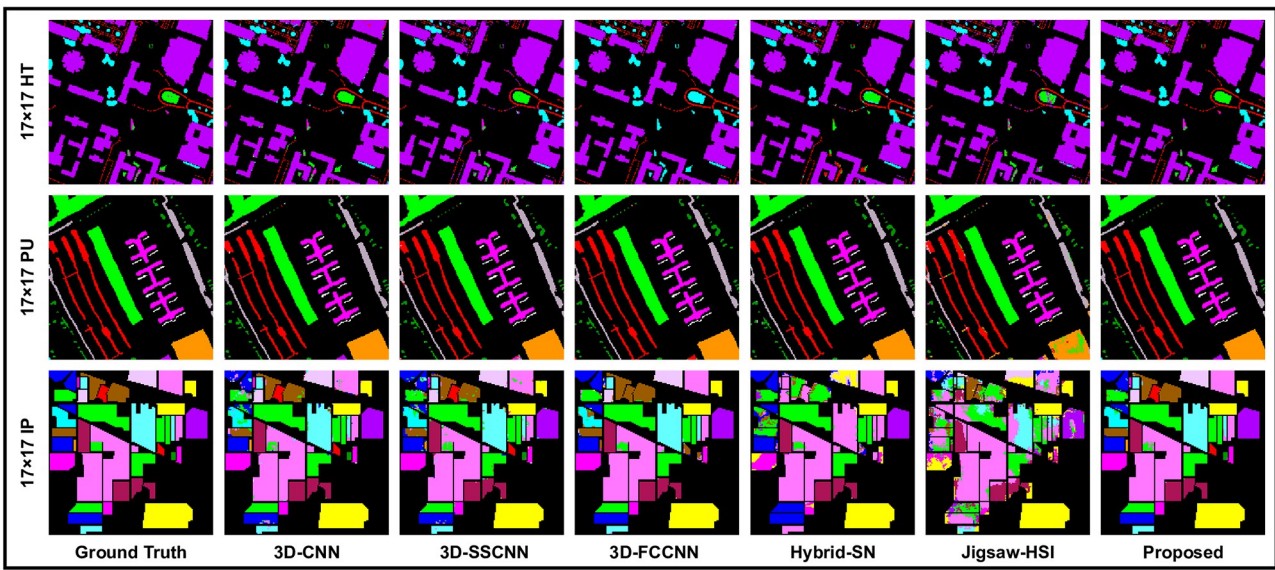

**Fig 7. Our models, which incorporate state-of-the-art techniques, perform sophisticated processing of the Ground Truths with high precision and accuracy, considering the complexity of each spatial dimension.**

Table 5. In light of the spatial dimension of the 21 × 21 window size, comparative evaluations were conducted with cutting-edge deep learning techniques.

| Datasets | HT | | | PU | | | IP | | |
|---|---|---|---|---|---|---|---|---|---|
| Models | OA | AA | Kappa | OA | AA | Kappa | OA | AA | Kappa |
| 3D-CNN | 96.77 | 72.80 | 88.36 | 99.17 | 77.57 | 98.96 | 96.36 | 84.74 | 95.84 |
| 3D-SSCNN | 96.41 | 67.87 | 88.34 | 99.49 | 88.49 | 99.36 | 95.63 | 83.07 | 95.01 |
| 3D-FCCNN | 97.50 | 59.99 | 91.46 | 99.32 | 77.28 | 99.15 | 97.12 | 73.44 | 96.17 |
| HYbrid-SN | 96.87 | 63.95 | 92.38 | 98.69 | 75.09 | 98.36 | 75.76 | 48.53 | 71.60 |
| JIgsaw-HSI | 95.20 | 58.06 | 82.34 | 95.16 | 76.89 | 93.93 | 73.04 | 51.11 | 68.67 |
| Proposed | **97.82** | **73.80** | **92.44** | **99.86** | **99.83** | **99.82** | **99.89** | **88.82** | **99.86** |

comparison line at 3rd place. Hybrid-SN and Jigsaw-HSI could not show the prominent classification prediction. On the other hand, the proposed 3D-CDCN Attention Dense Net consistently maintains the highest accuracy on all the evaluation metrics. These results can be seen in Table 6. In Fig 9, we can observe the classification maps of Houston 2018 (HT), Pavia University (PU), and Indian Pines(IP). These maps present the geographical characteristics of each class based on different window sizes (spatial dimensions).

Fig 10 illustrates the (loss, accuracy) across 100 epochs of training, comparing our method to other techniques. The results indicate that our method outperformed the alternatives in terms of both (loss, accuracy). The proposed method achieved good accuracy, demonstrating the efficiency of the proposed special attention mechanism. The spatial dimensions outlined in Table 6 were analyzed by our model, resulting in the computation of accuracy metrics such as (*OA*, *AA*, and *Kappa*). These metrics are presented in Table 2.

## 4.2 Convergence analysis

From Fig 10, it can be observed that the proposed special attention mechanism performs well and produces fast and accurate results in just 15 epochs. The proposed attention mechanism

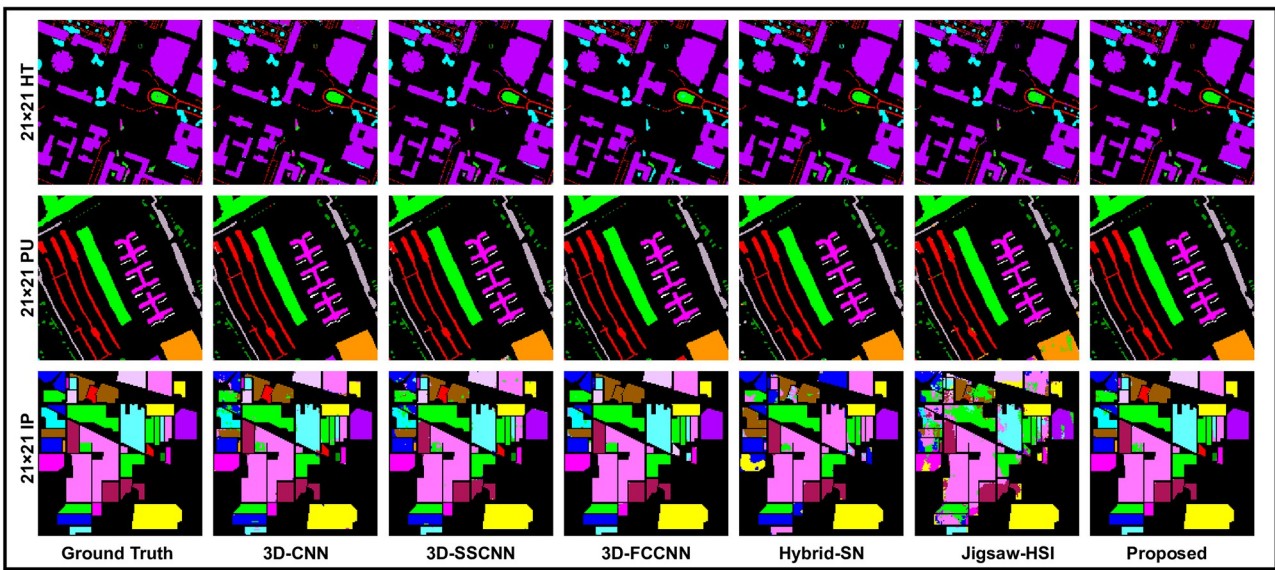

**Fig 8. Our models, which incorporate state-of-the-art techniques, perform sophisticated processing of the Ground Truths with high precision and accuracy, considering the complexity of each spatial dimension.**

**Table 6. Based on our research findings, the efficacy of the proposed model is contingent upon the 25 × 25 size of the window.**

| Datasets | HT | | | PU | | | IP | | |
|---|---|---|---|---|---|---|---|---|---|
| Models | OA | AA | Kappa | OA | AA | Kappa | OA | AA | Kappa |
| 3D-CNN | 97.46 | 66.96 | 91.07 | 99.17 | 77.57 | 98.96 | 97.75 | 95.35 | 97.43 |
| 3D-SSCNN | 96.56 | 72.01 | 88.84 | 98.86 | 87.70 | 98.58 | 96.62 | 95.06 | 96.14 |
| 3D-FCCNN | 97.55 | 60.14 | 91.60 | 99.30 | 77.21 | 99.13 | 96.73 | 73.10 | 96.27 |
| Hybrid-SN | 95.12 | 46.52 | 83.15 | 96.63 | 65.42 | 95.78 | 79.17 | 51.35 | 75.70 |
| Jigsaw-HSI | 94.78 | 58.60 | 80.33 | 96.86 | 79.96 | 96.07 | 82.39 | 65.41 | 79.68 |
| Proposed | **97.93** | **71.74** | **92.80** | **99.89** | **88.82** | **99.86** | **99.38** | **99.45** | **99.30** |

controls the height and width of the HSI between the blocks and produces robust results. The other factors that also participate in the fast convergence of the proposed model are listed below:

1. Selective Feature Learning: Attention mechanisms allow the model to selectively focus on specific regions or features within the input data. This selective attention enables the network to prioritize relevant information and ignore less important details, leading to a more efficient learning process.

2. Adaptive Learning Rates: Adam (short for Adaptive Moment Estimation) is an optimizer that maintains adaptive learning rates for each parameter. It computes the adaptive learning rates based on both the first-order moment (mean) and the second-order moment (uncentered variance) of the gradients. This adaptability allows Adam to converge quickly by adjusting the learning rates for each parameter individually.

3. Reduction of Redundant Information: Attention mechanisms help in identifying and emphasizing important spatial and temporal features in the 3D data. By reducing the

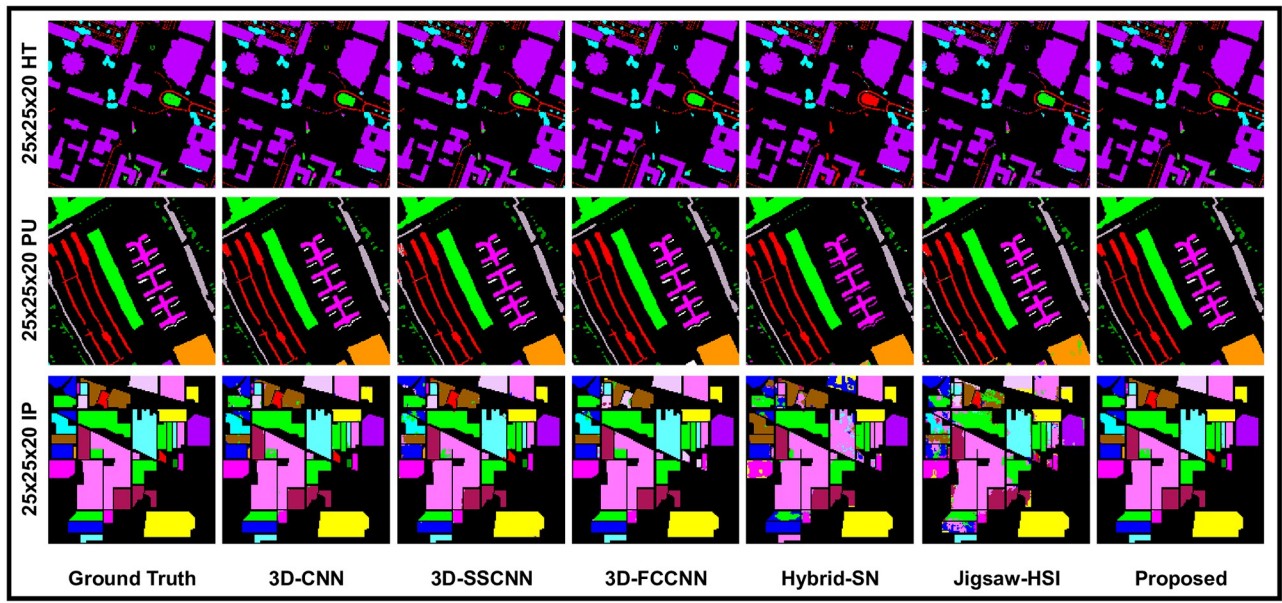

**Fig 9. Our models, which incorporate state-of-the-art techniques, perform sophisticated processing of the Ground Truths with high precision and accuracy, considering the complexity of each spatial dimension.**

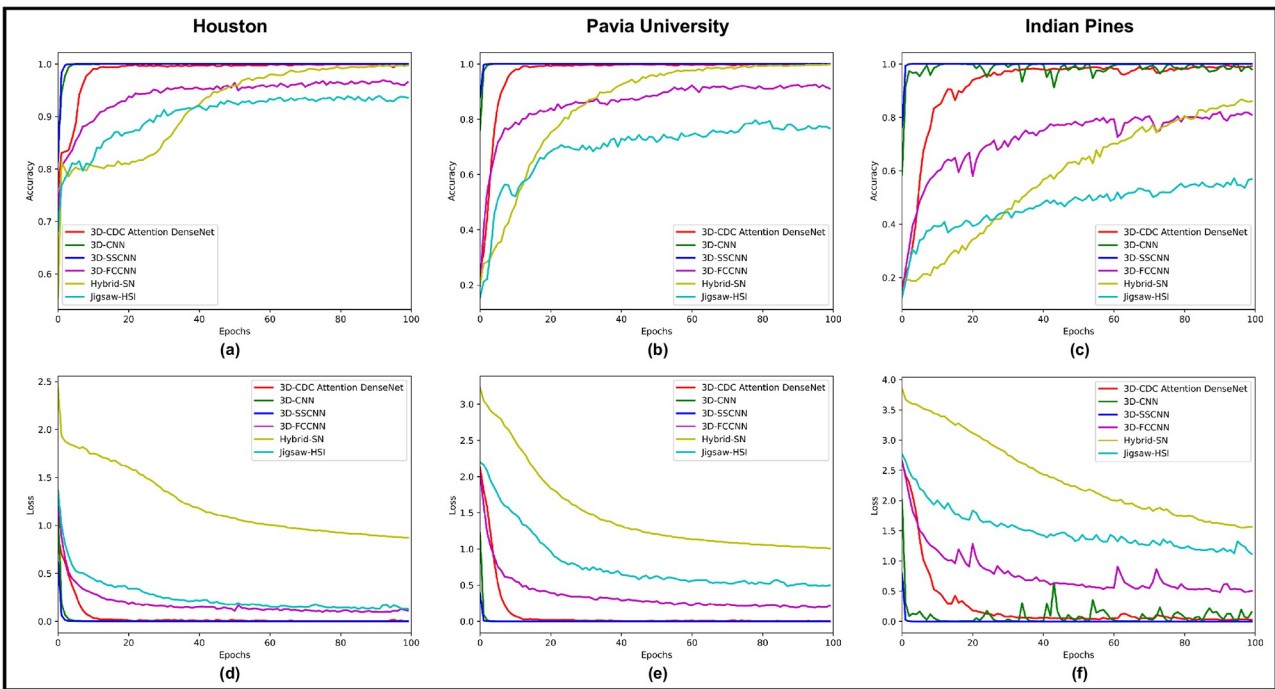

**Fig 10.** The epoch-wise evaluation of the Houston 2018, Pavia University and Indian Pines datasets, with a window patch size of 25 × 25, where a, b and c are the accuracy graphs, whereas d, e, and f are the loss graphs on 100 epochs.

emphasis on redundant or less informative regions, the model can converge faster as it learns to capture the most discriminative features.

4. Improved Gradient Flow: The attention mechanism can assist in improving the flow of gradients during backpropagation. By assigning higher gradients to more relevant features, the model can learn more effectively from the most critical information, facilitating faster convergence.

5. Enhanced Discriminative Power: Attention mechanisms enable the network to assign different weights to different input parts, allowing it to focus on the most discriminative features for the task at hand. This enhanced discriminative power can lead to quicker convergence as the model hones in on the crucial aspects of the data.

6. Facilitation of Long-Range Dependencies: In 3D data, capturing long-range dependencies is crucial for understanding temporal dynamics. Attention mechanisms facilitate the modeling of long-range dependencies by allowing the model to selectively attend to relevant frames or sequences, aiding faster convergence.

## 4.3 Ablation study

An ablation study was performed to measure the effectiveness of each module in the proposed architecture. 3D-CDC architecture is introduced with spatial attention and dense-net modules. We conducted ablation experiments, and the results are listed in Table 7. The first experiment was with CNN (vanilla convolution), and OA was achieved at 96.07%, and computational time

**Table 7. The ablation study was conducted in different scenarios and recorded for the (Houston 2018) with 25 × 25 window size of the experiment.**

| CNN | SA-CNN | CDCN | SA-CDCN | OA | Time |
|---|---|---|---|---|---|
| ✓ | ✗ | ✗ | ✗ | 96.07 | 2.78 |
| ✗ | ✓ | ✗ | ✗ | 96.27 | 3.98 |
| ✗ | ✗ | ✓ | ✗ | 97.88 | 4.98 |
| ✗ | ✗ | ✗ | ✓ | 97.93 | 4.38 |

was 2.78. In the second experiment, the efficiency of the spatial attention (SA) was measured with CNN. By adding the SA module, the over-efficacy of the CNN also increased with the OA 96.27% with 3.98 seconds. In the third ablation study, CDCN (center difference convolutional network) 97.88% OA using the 4.98 seconds. In the fourth ablation experiment, the effectiveness of 3D-CDCN with the SA module achieved the highest accuracy 97.93% with less time, i.e., 4.38 seconds, which is 0.6 seconds less than the network (CDCN) without the SA module. On the other hand, The SA module increases the performance of the proposed network.

## 4.4 Computational efficiency

Each model's performance is measured to determine the computational efficiency. The average run time is presented in Table 8, which shows that information is processed quickly with a small patch size, i.e., 13 × 13. HybridSN produced the results very fast and placed at the top. 3D-FCCNN is the second-best model to process the model, while Jigsaw-HSI is placed at 3rd in this series. Of course, our model is placed as the 4th best efficient model. The proposed method is deeper than the comparison algorithm. Our method handles the detailed intrinsic information and provides the best classification results. The proposed model becomes efficient when we go to the larger patch size of 25 × 25. 3DCDCN predicted the results just in 4.38 seconds, which is the second-best model on the larger size. While the other models cannot manage sustainability. So, we can say that the proposed model is the best fit and reduces the complexity of the HSIs.

## 4.5 Comparison with base-line deep learning methods

We have extensively evaluated the performance of the proposed approach, which is composed of a 3D layered architecture, by comparing it to several deep learning methods. At first, we examined our method in the context of 3D layered architecture during the experimental phase. Afterward, we expanded our research to include deep learning methods that expand beyond this specific architecture. The complete comparison of baseline deep learning models is shown in Table 9. The provided comparison table in this part clearly shows the superiority

**Table 8. The computational times measured in seconds were recorded for the HT experimental dataset across various window sizes.**

| Models | Window size | | | |
|---|---|---|---|---|
| | 13 × 13 | 17 × 17 | 21 × 21 | 25 × 25 |
| Proposed | 1.82 | 2.01 | 2.42 | 4.38 |
| Jigsaw-HSI | 1.67 | 1.95 | 5.85 | 9.45 |
| Hybrid-SN | 1.15 | 1.89 | 1.97 | 2.58 |
| 3D-FCCNN | 1.31 | 1.61 | 2.31 | 3.94 |
| 3D-SSCNN | 1.83 | 1.42 | 1.88 | 2.54 |
| 3D-CNN | 1.69 | 1.31 | 1.71 | 2.19 |

**Table 9. Comparison with base-line deep learning methods.**

| Datasets | HT | | | PU | | | IP | | |
|---|---|---|---|---|---|---|---|---|---|
| Models | OA | AA | Kappa | OA | AA | Kappa | OA | AA | Kappa |
| 1D CNN [85] | 80.23 | 62.09 | 78.61 | 75.39 | 83.13 | 68.73 | 73.19 | 83.29 | 69.52 |
| 2D-CNN [85] | 84.08 | 70.01 | 82.77 | 88.19 | 89.98 | 84.49 | 82.94 | 92.88 | 80.55 |
| SSFCN [86] | 84.14 | 69.84 | 82.86 | 85.99 | 90.32 | 81.83 | 89.62 | 94.25 | 88.12 |
| GCN [87] | 85.73 | 71.41 | 84.54 | 81.83 | 87.60 | 76.75 | 70.22 | 79.00 | 66.27 |
| Proposed | **97.93** | **71.74** | **92.80** | **99.89** | **88.82** | **99.86** | **99.38** | **99.45** | **99.30** |

of the approach we propose compared to common deep learning models such as 1D CNN, 2D-CNN, SSFCN, and GCN across three distinct datasets: HT, PU, and IP. The 1D Convolutional Neural Network (CNN) has a reasonable level of performance, with the maximum overall accuracy (OA) of 80.23% in the HT dataset. Its performance does, however, significantly decline in the PU and IP datasets, indicating there are limitations when handling these kinds of data. While 2D-CNN outperforms 1D CNN, particularly in the PU dataset with an overall accuracy (OA) of 88.19%, it is still not as effective as more advanced models. This implies that its ability to capture the complex structure of the datasets is limited. The SSFCN model consistently achieves good performance across all datasets, with its maximum overall accuracy (OA) of 89.62% observed in the IP dataset. This demonstrates a superior ability to manage diverse data structures in contrast to the prior models. GCN exhibits better performance on the HT dataset with an Overall Accuracy (OA) of 71.41%, its effectiveness significantly diminishes in the IP dataset, indicating possible challenges in extrapolating results across heterogeneous data types. The proposed method, demonstrates superior performance compared to all other models across all datasets, with outstanding Overall Accuracy (OA) scores of 97.93%, 99.89%, and 99.3% in the HT, PU, and IP datasets, respectively. The good results show the efficacy of the 3D layered architecture in managing diverse and intricate data structures. The significant superiority of our method over the baseline models emphasizes its improved capacity to reliably identify and predict results, even in complex scenarios.

The investigation highlights the improved efficiency of the proposed method in handling complex data structures and its superiority over conventional deep learning models. This comparison analysis not only confirms the success of our approach but also provides the way for its implementation in increasingly complex and diverse based on data environments.

## 4.6 Limitation of the network

Using 3D Convolutional Neural Networks (CNNs) with 3D attention helps handle moving images and time-related data. But, there are problems. These methods need much computing power because they process complex 3D data. This might slow down the process, especially with big sets of data. Also, there's a risk of the model getting too focused on small details in the training data. This might make it struggle when dealing with new or different data, especially if there isn't much training data available.

## 4.7 Discussion

Various experiments were performed to measure the effectiveness of the proposed 3D-CDCN attention-dense net. Comprehensive experiments were performed for this purpose. Three publicly available datasets, i.e., HT, PU, and IP, were taken for the experiments. The proposed method was tested on different patch sizes. First, the experiments were performed on the

proposed method using different patch sizes. For this purpose, the window size was set to 13 × 13, 17 × 17, 21 × 21 and 25 × 25 with 20 spectral bands, from the result Table 2 it is noticed that the proposed method's Performance increased with larger patch size.

In the second experiment, we compared the performance of the proposed method with the existing CNN-based methods for a fair comparison. Most methods are 3D, and an equal experimental environment is provided. These methods were also tested on different patch sizes. When we summarized all the experiments of comparing methods, the performance of these methods varied with different patch sizes and on different datasets. For example, the 3D-FCNN produced OA = 98.36% on the HT dataset, the second-best result. This model cannot maintain its position on the PU dataset, achieving OA = 95.21% and placed at the 3rd position, the same as with the IP dataset. Jigsaw-HSI produced the lowest results with window size 13 × 13. On the other side, when we see the performance of the proposed 3D-CDCN attention Dense Net with different window sizes, it shows the highest results and maintains its consistency.

The second discussion point is each method's efficiency and capacity. For instanceThe semi-supervised 3D-CNN can capture distinct spatial classes across various wavelengths, enabling the analysis of a broad spectrum of spectral data. Compared to conventional 2D-CNN, the Spectral-Spatial 3D-SSCNN introduces higher computational complexity and challenges obtaining a diverse and representative HSI dataset for training. However, the Fast and Compact 3D-FCCNN approach has a drawback. It divides the HSI cube into smaller overlapping patches, which leads to a loss of spatial context. Consequently, this method becomes unable to capture global relationships between different regions. The Hybrid-SN and Jigsaw-HSI consider the importance of interpretability and transparency losses. To address these challenges and improve HSI classification performance and robustness, the clever 3D-CDC Attention DenseNet has been developed. This model focuses on extracting spatial-spectral feature maps, utilizing joint local intrinsic detailed patterns and interrelation among spectral features. The attention mechanism and dense network incorporate low-rank frequency feature information and guide feature tuning. As a result, these advancements have successfully overcome the challenges faced by existing state-of-the-art models.

## 5 Conclusion

In this article, we proposed a method for HSI classification based on a center difference convolution approach that incorporates central difference into vanilla convolution to enhance its representational characteristics and improve its generalization capacity of the convolutional neural network to extract the detailed intrinsic features for the most accurate classification task. The proposed method exploits the 3D Attention mechanism to explore the more appropriate features, whereas 3D Central Difference CNN is used to extract the detailed intrinsic features, and the dense connections were employed to improve the robustness of the architecture. Although the 2D-CNN and 3D-CNN-based approaches have been widely used for the HSI classification, they have limitations, i.e., 2D CNN does not simultaneously analyze the spatial and spectral data. In contrast, the 3-D CNN emerges as a superior alternative because precise estimation of HSIC requires considering both spatial and spectral features, however, this method ignores the intrinsic features that are important for the accurate classification of HSIs. Our proposed algorithm achieves superior experimental outcomes on three HSI benchmark datasets HT-2018, PU, and IP, establishing state-of-the-art results with different window sizes i.e. 13 × 13, 17 × 17, 21 × 21 and 25 × 25 with 20 spectral bands, 3D-CDC Attention DenseNet, produced the better OA score against the comparing methods on all the three datasets, whereas 3D-CDC Attention DenseNet achieved the highest OA% on 25 × 25 × 20 window-sized patch

that shows the efficiency of the proposed method. Our experiments demonstrate that our approach not only surpasses conventional 3D CNN-based models but also shows superiority when compared with the baseline deep learning methods, i.e. ID, CNN, 2D CNN, SSFCN, and GCN, and outperforms state-of-the-art networks on various public benchmarks while maintaining lower complexity. For future work, we intend to introduce more convolutional operators with CNNs to make better generalizations of the CNN-based architectures for the HSI classification.

## Supporting information

**S1 File.**
(RAR)

## Author Contributions

**Conceptualization:** Mahmood Ashraf, Muhammad Umer, Shtwai Alsubai, Ala Abdulmajid Eshmawi.

**Data curation:** Mahmood Ashraf.

**Formal analysis:** Mahmood Ashraf, Ala Abdulmajid Eshmawi.

**Funding acquisition:** Raed Alharthi.

**Investigation:** Mahmood Ashraf, Ala Abdulmajid Eshmawi.

**Methodology:** Raed Alharthi, Muhammad Umer.

**Project administration:** Raed Alharthi.

**Resources:** Raed Alharthi.

**Software:** Raed Alharthi, Lihui Chen, Shtwai Alsubai.

**Supervision:** Lihui Chen, Muhammad Umer.

**Validation:** Lihui Chen.

**Visualization:** Lihui Chen, Muhammad Umer, Shtwai Alsubai.

**Writing – original draft:** Shtwai Alsubai.

**Writing – review & editing:** Muhammad Umer, Ala Abdulmajid Eshmawi.

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
