## [Decision Letter · Decision Letter 0]

16 Nov 2023

PONE-D-23-32380Attention 3D Central Difference Convolutional Dense Network for Hyperspectral Image ClassificationPLOS ONE

Dear Dr. Umer,

Thank you for submitting your manuscript to PLOS ONE. After careful consideration, we feel that it has merit but does not fully meet PLOS ONE’s publication criteria as it currently stands. Therefore, we invite you to submit a revised version of the manuscript that addresses the points raised during the review process.

We look forward to receiving your revised manuscript.

Kind regards,

Nouman Ali

Academic Editor

PLOS ONE

4. We note that Figure 5 and 7 in your submission contain copyrighted images. All PLOS content is published under the Creative Commons Attribution License (CC BY 4.0), which means that the manuscript, images, and Supporting Information files will be freely available online, and any third party is permitted to access, download, copy, distribute, and use these materials in any way, even commercially, with proper attribution. For more information, see our copyright guidelines: http://journals.plos.org/plosone/s/licenses-and-copyright.

a. You may seek permission from the original copyright holder of Figure 5 and 7 to publish the content specifically under the CC BY 4.0 license. 

Reviewers' comments:

Reviewer's Responses to Questions

**Comments to the Author**

1. Is the manuscript technically sound, and do the data support the conclusions?

Reviewer #1: Yes

Reviewer #2: Yes

Reviewer #3: Yes

Reviewer #4: Partly

2. Has the statistical analysis been performed appropriately and rigorously? 

Reviewer #1: No

Reviewer #2: Yes

Reviewer #3: I Don't Know

Reviewer #4: No

3. Have the authors made all data underlying the findings in their manuscript fully available?

Reviewer #1: Yes

Reviewer #2: Yes

Reviewer #3: No

Reviewer #4: Yes

4. Is the manuscript presented in an intelligible fashion and written in standard English?

Reviewer #1: Yes

Reviewer #2: Yes

Reviewer #3: No

Reviewer #4: Yes

5. Review Comments to the Author

Reviewer #1: Your research addresses an important issue, and your proposed algorithm shows promise. However, there are several areas that could be improved to enhance the clarity and rigor of your work. Here are the specific suggestions for revision:

1. This article lacks a background introduction to such massive data. Writing background refers to the historical background in which the author created this article, and the context in which it was created. Therefore, please optimize the writing background of the article so that it can have a deeper understanding of the content.

2. In the introduction section of the article, the author's expression of the writing meaning is somewhat vague and difficult to understand. Please reorganize the language to express it so that readers can quickly understand the writing meaning and purpose of the article.

3. This article introduces the research of multiple scholars and explains their research methods. However, the literature section can emphasize the shortcomings of various methods, which is enough to shift to newly proposed methods.

4. Provide information on the specific data used in your study. Mention the source and characteristics of the data.

6. Describe the simulation environment and parameters used for testing the algorithm.

7. Include a discussion on the metrics used to evaluate the algorithm's performance and why these metrics were chosen.

8. Provide a more detailed presentation of the simulation results, including tables or graphs to illustrate the improvements achieved.

9. Discuss the computational complexity of your proposed algorithm and how it compares to existing methods.

10. For evaluation purposes, the text should include a discussion section that compares the results with existing literature.

11. What are the limitations of this study?

12. The conclusion of this paper still provides a lot of background information, which obviously does not meet the requirements of conclusion writing. In addition, this article did not explain the shortcomings of the experimental section and the direction of future research.

Reviewer #2: This paper proposed Attention 3D Central Difference Convolutional Dense Network for Hyperspectral Image Classification. Overall, the structure of this paper is well organized, and the presentation is clear. However, there are still some crucial problems that need to be carefully addressed before a possible publication. More specifically,

1. A deep literature review should be given, particularly advanced and SOTA deep learning or AI models in remote sensing. 

2. Please clarify the contributions, and what are the newly-added values compared to the existing methods.

3. The workflow should be drawn to clarify the proposed method.

4. The ablation analysis should show the performance gain from newly-added modules.

5. How about the computational complexity?

6. It is well-known that the remote sensing data usually tend to suffer from various degradation, noise effects, or variabilities in the process of imaging. Please give the discussion and analysis by referring to the paper titled by e.g., An Augmented Linear Mixing Model to Address Spectral Variability for hyperspectral unmixing. The reviewer is wondering what will happen if the proposed method meets the various variabilities.

7. Some future points should be pointed out in the conclusion.

Reviewer #3: This article introduces a novel approach called the Attention 3D Central Difference Convolutional Dense Network. It uses the 3D-Central Difference Convolutional method to handle local intrinsic patterns in the spatial-spectral feature maps. This approach incorporates pixel-level concatenation and spatial attention mechanism within a dense strategy to include low-rank frequency features and guide feature turning. Although this idea is helpful for Hyperspectral image classification, there are still some issues that need to be addressed.

1、Please provide the abbreviations that appear in the manuscript along with their corresponding full forms. For example, 3D-CDC.

2、Please adjust the placement of the citation in the text, for example, in line 60.

3、Please specify the content corresponding to each chapter before the Proposed Methodology.

4、Please provide the figures mentioned in the manuscript at the title position of the figures section.

5、Please provide the standardized formatting for Table 1, Table 2, Table 3, Table 4, and Table5 in the manuscript.

6、Please standardize the font size of the title in the manuscript.

7、Please provide a clear description of the Central Difference Convolution module mentioned in the manuscript.

8、Please further expand the experiment content of the network based on the given experiment in the manuscript (e.g., the computational complexity of the network) to fully demonstrate whether the proposed solution in the manuscript is reasonable.

Reviewer #4: The Authors proposed 3D difference Dense NET framework for HSI classification. The performance of the proposed framework experimented on the benchmark datasets . However ,the following comments need to address before manuscript accept for publication .

1. Novelty and contribution needs to specify properly .

2. How proposed framework differ from the existing 3DCNN CDC framework ?

3. Literature review weak. Authors needs to study the recent existing 3DCNN CDC models for HSI classification and Spectral spatial features based HSI classification frameworks and same needs to compare the performance of the proposed .

4. convergence analysis is missing . Is the parameters used in the framework generalize ?

5. Justifications for fast convergence is required.

6. PLOS authors have the option to publish the peer review history of their article (what does this mean?). If published, this will include your full peer review and any attached files.

Reviewer #1: **Yes: **Alireza Sharifi

Reviewer #2: No

Reviewer #3: No

Reviewer #4: **Yes: **Prabukumar Manoharan

---

## [Author Response · Author response to Decision Letter 0]

9 Dec 2023

We have provided a separate response to reviewer file.

---

## [Decision Letter · Decision Letter 1]

26 Jan 2024

PONE-D-23-32380R1Attention 3D Central Difference Convolutional Dense Network for Hyperspectral Image ClassificationPLOS ONE

Dear Dr. Umer,

Thank you for submitting your manuscript to PLOS ONE. After careful consideration, we feel that it has merit but does not fully meet PLOS ONE’s publication criteria as it currently stands. Therefore, we invite you to submit a revised version of the manuscript that addresses the points raised during the review process.

We look forward to receiving your revised manuscript.

Kind regards,

Nouman Ali

Academic Editor

PLOS ONE

Reviewers' comments:

Reviewer's Responses to Questions

**Comments to the Author**

1. If the authors have adequately addressed your comments raised in a previous round of review and you feel that this manuscript is now acceptable for publication, you may indicate that here to bypass the “Comments to the Author” section, enter your conflict of interest statement in the “Confidential to Editor” section, and submit your "Accept" recommendation.

Reviewer #1: (No Response)

Reviewer #2: (No Response)

Reviewer #4: All comments have been addressed

Reviewer #5: (No Response)

Reviewer #6: (No Response)

2. Is the manuscript technically sound, and do the data support the conclusions?

Reviewer #1: Yes

Reviewer #2: (No Response)

Reviewer #4: Yes

Reviewer #5: Yes

Reviewer #6: Yes

3. Has the statistical analysis been performed appropriately and rigorously? 

Reviewer #1: No

Reviewer #2: (No Response)

Reviewer #4: Yes

Reviewer #5: No

Reviewer #6: Yes

4. Have the authors made all data underlying the findings in their manuscript fully available?

Reviewer #1: No

Reviewer #2: (No Response)

Reviewer #4: Yes

Reviewer #5: Yes

Reviewer #6: Yes

5. Is the manuscript presented in an intelligible fashion and written in standard English?

Reviewer #1: Yes

Reviewer #2: (No Response)

Reviewer #4: Yes

Reviewer #5: No

Reviewer #6: Yes

6. Review Comments to the Author

Reviewer #1: Your research addresses an important issue, and your proposed algorithm shows promise. However, there are several areas that could be improved to enhance the clarity and rigor of your work. Here are the specific suggestions for revision:

1. This article lacks a background introduction to such massive data. Writing background refers to the historical background in which the author created this article, and the context in which it was created. Therefore, please optimize the writing background of the article so that it can have a deeper understanding of the content.

2. In the introduction section of the article, the author's expression of the writing meaning is somewhat vague and difficult to understand. Please reorganize the language to express it so that readers can quickly understand the writing meaning and purpose of the article.

3. This article introduces the research of multiple scholars and explains their research methods. However, the literature section can emphasize the shortcomings of various methods, which is enough to shift to newly proposed methods. It is better to add the following references to enrich the work:

4. Provide information on the specific data used in your study. Mention the source and characteristics of the data.

6. Describe the simulation environment and parameters used for testing the algorithm.

7. Include a discussion on the metrics used to evaluate the algorithm's performance and why these metrics were chosen.

8. Provide a more detailed presentation of the simulation results, including tables or graphs to illustrate the improvements achieved.

9. Discuss the computational complexity of your proposed algorithm and how it compares to existing methods.

10. For evaluation purposes, the text should include a discussion section that compares the results with existing literature.

11. What are the limitations of this study?

12. The conclusion of this paper still provides a lot of background information, which obviously does not meet the requirements of conclusion writing. In addition, this article did not explain the shortcomings of the experimental section and the direction of future research.

Reviewer #2: The reviewer has saw the authors‘ efforts. The reviewer only has the following comments:

1. Several current advanced AI methods should be further discussed and analyized in remote sensing, e.g., SpectralGPT, 10.1109/TGRS.2023.3279834, 10.1016/j.rse.2023.113856

2. The reviewer is also wondering about the computational complexity.

Reviewer #4: The authors addressed all my previous comments, I recommend the revised manuscript for possible publication.

Reviewer #5: Hyperspectral Images (HSI) classification is a challenging task due to a large number of spatial-spectral bands of images with high inter-similarity, extra variability classes, and complex region relationships, including overlapping and nested regions. Classification becomes a complex problem in remote sensing images like HSIs. Convolutional Neural Networks (CNNs) have gained popularity in addressing this challenge by focusing on HSI data classification. However, the performance of 2D CNN methods heavily relies on spatial information, while 3D-CNN methods offer an alternative approach by considering both spectral and spatial information. Nonetheless, the computational complexity of 3D-CNN methods increases significantly due to the large capacity size and spectral dimensions. These methods also face difficulties in manipulating information from local intrinsic detailed patterns of feature maps and low rank frequency feature tuning. To overcome these challenges and improve HSI classification performance, authors propose an innovative approach called the Attention 3D Central Difference Convolutional Dense Network (3D-CDC Attention DenseNet). Our 3D-CDC method leverages the manipulation of local intrinsic detailed patterns in the spatial-spectral features maps, utilizing pixel-wise concatenation and spatial attention mechanism within a dense strategy to incorporate low-rank frequency features and guide the feature tuning. Experimental results on benchmark datasets such as Pavia University, Houston 2018, and Indian Pines demonstrate the superiority of our method compared to other HSI classification methods, including state-of-the-art techniques. Authors must address my following queries

1. Comparison with base-line deep learning method must be presented.

2. The proposed research must be evaluated on a large scale dataset otherwise mentioned a reason.

3. The mathematical model is not defined well, a detailed explanation is required.

4. The details about fine-tunned parameters/optimization details are missing.

Reviewer #6: The paper written by Ashraf et al. is very important for the research community. The structure of the paper is good, results are clearly explained with proper citations to explain the paper. I have some minor comments to improve the quality of the paper. The minor comments are:

1. Discourage the usage of words 'we', 'our', and 'they'.

2. Authors need to add a result line in the Abstract.

3. Please confirm Figure 1 you drew or add a proper citation to it.

4. There are some typos in the paper. Authors need to carefully re-write the long sentences of the paper.

7. PLOS authors have the option to publish the peer review history of their article (what does this mean?). If published, this will include your full peer review and any attached files.

Reviewer #1: No

Reviewer #2: No

Reviewer #4: No

Reviewer #5: No

Reviewer #6: No

---

## [Author Response · Author response to Decision Letter 1]

14 Feb 2024

We have provided a separate PDF file to give response to reviewer comments.

---

## [Decision Letter · Decision Letter 2]

20 Feb 2024

Attention 3D Central Difference Convolutional Dense Network for Hyperspectral Image Classification

PONE-D-23-32380R2

Dear Dr. Umer,

We’re pleased to inform you that your manuscript has been judged scientifically suitable for publication and will be formally accepted for publication once it meets all outstanding technical requirements.

Kind regards,

Nouman Ali

Academic Editor

PLOS ONE

Additional Editor Comments (optional):

Reviewers' comments:

Reviewer's Responses to Questions

**Comments to the Author**

1. If the authors have adequately addressed your comments raised in a previous round of review and you feel that this manuscript is now acceptable for publication, you may indicate that here to bypass the “Comments to the Author” section, enter your conflict of interest statement in the “Confidential to Editor” section, and submit your "Accept" recommendation.

Reviewer #2: All comments have been addressed

Reviewer #5: All comments have been addressed

Reviewer #6: (No Response)

2. Is the manuscript technically sound, and do the data support the conclusions?

Reviewer #2: Yes

Reviewer #5: Yes

Reviewer #6: (No Response)

3. Has the statistical analysis been performed appropriately and rigorously? 

Reviewer #2: Yes

Reviewer #5: Yes

Reviewer #6: (No Response)

4. Have the authors made all data underlying the findings in their manuscript fully available?

Reviewer #2: (No Response)

Reviewer #5: Yes

Reviewer #6: (No Response)

5. Is the manuscript presented in an intelligible fashion and written in standard English?

Reviewer #2: No

Reviewer #5: Yes

Reviewer #6: (No Response)

6. Review Comments to the Author

Reviewer #2: It is ready to accept the current version. The comments from the reviewer have been addressed. The reviewer dose not have more comments.

Reviewer #5: The quality of this manuscript is now up-to-the-mark and i will suggest the editor to accept this manuscript.

Reviewer #6: (No Response)

7. PLOS authors have the option to publish the peer review history of their article (what does this mean?). If published, this will include your full peer review and any attached files.

Reviewer #2: No

Reviewer #5: No

Reviewer #6: No

---

## [Editor Report · Acceptance letter]

26 Mar 2024

PONE-D-23-32380R2 

PLOS ONE

Dear Dr. Umer, 

I'm pleased to inform you that your manuscript has been deemed suitable for publication in PLOS ONE. Congratulations! Your manuscript is now being handed over to our production team.

Kind regards, 

on behalf of

Dr. Nouman Ali 

Academic Editor

PLOS ONE